# Profiles of Loneliness and Social Isolation in Physically Active and Inactive Older Adults in Rural England

**DOI:** 10.3390/ijerph18083971

**Published:** 2021-04-09

**Authors:** Jolanthe de Koning, Suzanne H Richards, Grace E R Wood, Afroditi Stathi

**Affiliations:** 1Department for Health, University of Bath, Bath BA2 7AY, UK; dekoning.jolanthe@gmail.com; 2Leeds Institute of Health Sciences, University of Leeds, Leeds LS2 9NL, UK; s.h.richards@leeds.ac.uk; 3School of Sport, Exercise and Rehabilitation Sciences, University of Birmingham, Birmingham B15 2TT, UK; GXW877@student.bham.ac.uk

**Keywords:** aging, mental health, qualitative, activity, socio-ecological, environment

## Abstract

*Objective:* Loneliness and social isolation are associated with higher risk of morbidity and mortality and physical inactivity in older age. This study explored the socioecological context in which both physically active and inactive older adults experience loneliness and/or social isolation in a UK rural setting. *Design:* A mixed-methods design employed semi structured interviews and accelerometer-measured moderate-to-vigorous physical activity (MVPA). Interviews explored the personal, social and environmental factors influencing engagement with physical activities, guided by an adapted-socioecological model of physical activity behaviour. *Findings:* Twenty-four older adults (Mean Age = 73 (5.8 SD); 12 women) were interviewed. Transcripts were thematically analysed and seven profiles of physical activity, social isolation and loneliness were identified. The high-MVPA group had established PA habits, reported several sources of social contact and evaluated their physical environment as activity friendly. The low MVPA group had diverse experiences of past engagement in social activities. Similar to the high MVPA, they reported a range of sources of social contact but they did not perceive the physical environment as activity friendly. *Conclusions:* Loneliness and/or social isolation was reported by both physically active and inactive older adults. There is wide diversity and complexity in types and intensity of PA, loneliness and social isolation profiles and personal, social and environmental contexts.

## 1. Introduction

Large cohort studies and meta-analyses highlight that loneliness and social isolation are independently associated with increased risk of morbidity and mortality in older age [1,2,3,4]. Predictors and triggers of loneliness and social isolation for older people living in rural areas of the UK are less studied than their urban counterparts despite greatly differing environmental and social contexts [5,6].

Perlman and Peplau [7] defined loneliness as “the unpleasant experience that occurs when a person’s network of social relations is deficient in some important way, either quantitatively or qualitatively” (p. 31). Two theories of loneliness contend that a mismatch between expectations and realities of social contact (Cognitive Theory) [7] and the deficit of intimate or social relationships (Deficit Theory) [8] can lead to feelings social or emotional loneliness.

Social isolation has been defined as lack of weekly direct contact with family, friends or neighbours [9] which may not necessarily be a negative experience [10]. It is widely recognised that loneliness and social isolation are associated but conceptually and empirically distinct [9,10,11,12,13]. It could be argued that frequent social contact is inherently active as it requires people to go outside their own house, and therefore reduced contacts and subsequent social isolation could be associated with lower PA levels. However, there are inactive forms of social contact, which could allow someone to interact socially but be physically inactive. This has been evident during the COVID-19 pandemic where many older adults maintained their social interaction via the increased use of digital technologies.

Qualitative studies have not always distinguished loneliness from social isolation. A content analysis of interviews with 30 adults (85 to 103 years) who lived alone [14], qualitative interviews with 12 Danish adults (70 and 79 years) [15] and a phenomenological study of 26 participants (12 to 82 years) [16] each described a “positive”/“empowering” form of loneliness, next to a loneliness that was “wrong”/“ugly”. In these studies, the positive experiences of “loneliness” may also be interpreted as situations in which participants were socially isolated, but not lonely, as they did not gain an unpleasant experience from it, which is not uncommon in older adults living in rural areas [10].

People experiencing social isolation or loneliness are more likely to have other risk factors, such as socioeconomic adversity and unhealthy lifestyles, that predict negative health outcomes. [1]. Targeting unhealthy lifestyles, physical inactivity in particular, with policies and public health interventions could potentially reduce excess mortality among people who are isolated and/or lonely. Numerous studies support the association of loneliness and social isolation with self-reported physical inactivity [17,18,19,20,21,22]. In contrast studies using objective measures of PA (accelerometery-measured PA) do not support this association [23,24,25]. The contradictory evidence might stem from people’s ability to recall more accurately information about their involvement in specific exercise pursuits rather than information about daily activity in general. which in turn may relate in different ways to experiences of social isolation and/or loneliness [26].

Despite the substantial body of qualitative literature linking social support and positive social experiences to levels of PA in older age, there is limited evidence about experiences of social isolation or loneliness in both active and inactive older adults. A systematic review of 37 studies by Pels and Kleinert [20] identified only one qualitative study which included aspects of loneliness and PA, and this study focussed on people with schizophrenia [27] which may not be transferrable to the wider populations of older people.

The aim of this study was to explore the personal, social and environmental context in which physically active and physically inactive older adults, living in a rural setting, experience loneliness and/or social isolation.

## 2. Materials and Methods

### 2.1. Study Design and Theoretical Framework

This study employed a qualitative methods design, with quantitative data collected previously in the Staying Healthy and Active in Rural Places (SHARP) study [6] and used for purposefully selecting participants for the qualitative study and to assist in data analysis Doyle, Brady [28]. The SHARP cross-sectional study explored the association between device-measured PA and specific activities out of the house with loneliness or SI from friends, family or neighbours. One-hundred-and-twelve participants (mean age = 72.8 [SD = 6.6], 51.8% female) from 23 villages in rural England completed questionnaires, seven-day accelerometery and activity diaries. Waist-mounted Actigraph (GT3X) accelerometers were worn for seven consecutive days during waking hours (not in water), as done previously with older adults [24,29,30,31]. Data were extracted using the Actilife v6.11.2 software, and considered valid when available for at least 10 h/day on five or more days. Three ordinal PA variables were computed: light PA (LPA), moderate-to-vigorous PA (MVPA) and total PA (TPA) using the Freedson, Melanson, and Sirard adult cut-off values [32]. Loneliness was assessed with a single-item measure of loneliness [33] and social isolation was assessed with three variables constructed using questions from the Social Capital Module (SCM) [34]: “How often do you meet up with relatives who are not living with you?”; “How often do you meet up with friends?”; and “How often do you speak to neighbours face-to-face?” These three questions were assessed independently leading to the creation of three variables: “SI from family”; “SI from friends”; and “SI from neighbours”.

For the qualitative study, an adapted-socioecological model of PA behaviour over the life-course [35] was employed to explore current and past personal, social and environmental factors associated with experiences of social isolation and/or loneliness in older rurally living adults. High and low levels of activity were expressed as quintiles of moderate-to-vigorous physical activity (MVPA). The socioecological model of health-behaviour recognises that intrapersonal, interpersonal/cultural, organizational, physical environment, and policy factors influence human health-related behaviour both directly and through interactions between these levels of influence [36,37].

In this study social isolation was defined as “less than weekly direct contact with family, friends and neighbours” [9] (p. 2). The cognitive theory of loneliness [7] and the deficit theory of loneliness, which includes the lack of intimate relationships (emotional loneliness) and the lack of wider social relationships (social loneliness) [8], were used to identify experiences of loneliness.

### 2.2. Participant Selection and Recruitment

Participants were purposefully recruited from the SHARP study population sample of 112 adults aged 65 and above, living across rural villages in South-West England [23]. The original sample was recruited through NHS general practice lists. The study was approved by the London-Central NHS Research Ethics Committee (reference number: 14/LO/0456) and the local NHS Research & Development committee (Reference: 2014/008). Participants were selected based on PA levels (seven-day accelerometer-measured PA), gender, age and physical function (Short Physical Performance Battery) [38]. MVPA was chosen as the indicator for PA level because the attainment of at least 30 min of MVPA per day has been shown to have a preventative and therapeutic effect on physical and mental health in older age [39]. Participants gave informed consent at the start of the quantitative phase of the SHARP study to be invited to contribute to a qualitative interview.

The quantitative sample (*n* = 112) was divided into quintiles of mean daily minutes of MVPA. Ten participants from the lowest MVPA quintile (0.6 to 8.5 min/day) and 10 from the highest MVPA quintile (53.3 to 113.1 min/day) were identified with the objective of matching the distributions of sex, age and physical function scores in both groups. Adequate matching of ages and physical function scores was not possible using only the lowest and highest MVPA quintiles. Therefore, two additional participants were selected from the second from lowest (8.6 to 18.9 min/day) and second from highest MVPA quintiles (39.1 to 52.5 min/day) to balance the groups. The identified participants were invited via a phone call by the first author, and all agreed to take part. The final sample of 24 participants residing in 20 villages (out of the 23 villages included in the SHARP study) was deemed sufficient as it ensured variety in age, gender and physical function scores in both highly active and inactive groups.

### 2.3. Data Collection

Twenty-four semi-structured interviews (45 to 90 min) were conducted and audio recorded by JdeK in the participants’ own homes. Interviews took place between three and four months after the quantitative data collection of the SHARP study. Before commencing, each participant signed a consent form.

An interview guide was used to explore the personal, social and environmental factors influencing participants’ engagement with physical activities (from earlier life to present), as well as the effect of their physical activities on their wellbeing (see full interview guide in Appendix A). A 7-day activity diary which had been completed in the quantitative phase of the SHARP study was used to prompt participants about the types of activities they undertook. The interview guide was piloted with the first three participants and amended to increase the clarity of questions.

Loneliness and social isolation were not specifically mentioned in an attempt to avoid distress or leading participants’ responses. However, when personal and social themes came up, or if loneliness, or indications of dissatisfaction with the level of social contact, were mentioned, then the researcher probed deeper to gain an understanding about whether the participants were isolated or felt lonely. Field notes were made straight after the interviews and raw thematic observations were discussed regularly with AS and SHR.

During the interviews every effort was made to not use ageist ideologies or language such as assuming a decreased level of PA, social interaction or wellbeing in older age, or using derogative words such as “elderly” [40].

### 2.4. Data Analysis

Interview audio recordings were transcribed and content analysis was performed using *NVivo 10.2.1* software (QSR International, Chadstone, VIC, Australia). JdeK completed deductive and inductive analysis identifying themes that did or did not fit into the adapted socioecological model and framework. To assure qualitative rigour, the lead author was responsive to emerging themes during data collection [41]. To minimise personal influences, the coding structure was examined by JdeK, AS and SHR until consensus was reached [42]. Data coding and interpretation were conducted in three stages.

#### 2.4.1. Stage 1. Coding of Qualitative Data

The first author created raw interview codes using the pre-specified themes (personal, social and environmental influences) of the adapted socioecological model ensuring the reporting of a lifetime approach distinguishing between influences from earlier life to present day. The raw code structure was then grouped into lower- and higher-order themes.

#### 2.4.2. Stage 2. Classification of MVPA, Social Isolation and Loneliness Profiles

Accelerometer, questionnaire and interview data were triangulated to classify participants into MVPA, social isolation and loneliness profiles enhancing the trustworthiness of the created profiles [43]. The questionnaire responses to social isolation and loneliness items were used as rough indicators of loneliness and social isolation, but qualitative data were given more weight, due to the timely person- and context-specific information [44,45]. Furthermore, as the questionnaires had been administered three to six months prior to the interviews, the possibility of changes in social isolation or loneliness status between the two data collection points could not be ruled out.

Participants were classified into MVPA and questionnaire-measured social isolation and loneliness profiles. Given the low prevalence of social isolation (based on the classification of less than weekly contact with friends, family and neighbours) in the SHARP study (8/112, 7.1%) [23], the threshold was lowered to having less than weekly contact with at least two of friends, family or neighbours.

Next, the loneliness and social isolation codes derived from the qualitative data were re-read and interpreted for evidence supporting or contradicting the classifications of loneliness or social isolation from the questionnaire data. Certain types of statements were interpreted into social isolation and types of loneliness (Table 1). As the social stigma of loneliness has been argued to stop individuals openly characterising themselves as lonely [46,47], the definitions of loneliness based in the deficit and cognitive theories of loneliness were used to identify loneliness, even if the word “lonely” was not mentioned.

The interpretation of qualitative data was double-checked by the corresponding author, an experienced qualitative researcher, to allow a process of critical dialogue and reflexivity [48].

#### 2.4.3. Stage 3. Interpretation of Socioecological Factors Relating to Profiles

The coded themes were re-read and summarised for each individual. The individual summaries were compared across all participants in each MVPA, social isolation and loneliness profile, and the common themes were extracted and presented in the findings with illustrative quotations.

## 3. Results

### 3.1. Participant Characteristics

Twenty-four older adults between 66 and 87 years of age with MVPA ranging from two to 113 min/day) were recruited (see Table 2 for sample characteristics). The low-MVPA group had a higher mean age (75.3 versus 70.8 years) and lower mean physical function score (8.7/12 versus 10.3/12) than the high-MVPA group. Education level was lower, widowhood more prevalent, and income lower in the low-MVPA group versus the high-MVPA group.

### 3.2. Profiles of Loneliness, Social Isolation and Physical Activity

Seven profiles of PA, social isolation and loneliness were identified: A to G (Table 3).

Most profiles of PA, loneliness and social isolation were characterised by different personal and social characteristics, while environmental circumstances differed only between high- and low-MVPA while the theme of “car dependence” was relevant across all profiles (Figure 1).

#### 3.2.1. High-MVPA Group

##### Personal Domain

Most older adults in the high-MVPA group (regardless of social isolation or loneliness) had established PA habits which aided them to continue engaging in active pursuits in older age. The extent to which past active pursuits had been socially orientated differed. Those who were not socially isolated and not lonely (Profile A) had a history of engaging in both social and active pursuits such as team sports and social committees. Those who were socially isolated but physically active (Profile C) had life-long interests based on productive pursuits such as gardening and coaching athletics. The physically active individuals reporting emotional loneliness (Profile E) had a history of engaging in exercise pursuits such as dog walking, swimming or group exercise classes in which no close friends were made.


*“Yes I mean I’ve played tennis, ever since I was at school, so… sport and music actually, is the two things they [the school] excelled in, and I think that’s probably had a fair, um, influence on my life”*
(Vanessa, Profile A, age 71, 65 min. MVPA/day)


*“I started coaching [athletics]… I started that and um yes, so [age] 55, 56, I’m 74, yea, so it’s continuing”; “I have always been a member of athletics clubs. Right from the age of 14 or something.”*
(Robert, Profile C, age 74, 58 min. MVPA/day)


*“I’ve always been active dog walking and swimming”; “Even if I don’t feel like It I do it”*
(Margery, Profile E, age 70, 113 min. MVPA/day)

The motives for active pursuits also differed. Those who were not socially isolated or lonely (Profile A) were motivated by social contact to engage in PA, exercise or sports activities. Those who were socially isolated (Profile C) were mainly motivated by knowledge seeking and sharing, and those experiencing emotional loneliness (Profile E) were mainly motivated by health-reasons and enjoyment of being active.


*“Very much the social side, it’s very strong. Um, I wouldn’t go out and play golf on my own for example. You see one or two guys doing that but um, it’s… the fun of going.”*
(Phil, Profile A, age 66, 89 min. MVPA/day)


*“When I go on my London [seed merchant] shows, you know, you have to be quite knowledgeable because people ask you quite searching questions. I quite like questions that I don’t know the answer to because I can go home and look them up.”*
(Rose, Profile C, age 68, 84 min. MVPA/day)


*“I’m actually very aware of keeping very physically fit because if you’re not very physically fit you’re very restricted in what you can do.”*
(Barbara, Profile E, age 68, 66 min. MVPA/day)

##### Social Domain

The sources of social contact differed across the high-MVPA group. A lack of contact with family was only experienced by participants who experienced emotional loneliness (Profile E). Their discontent with social contact highlighted past family relationship disruption, leading them to feel sad or depressed at present, even though they had frequent contact with neighbours and local/further friends.


*“Probably one of the biggest things is I’ve got two sons and my youngest son has fallen out with the rest of the family. And um, that’s where my two grandchildren are, so I never see them… So, I feel I’ve missed out… it is quite um, I get quite distressed… quite depressed about it at times.”*
(Barbara, age 68, 66 min. MVPA/day)


*“I was more active when I was at [age] 40 s to 50 s because, and then my partner walked out on me and broke my heart and life went down-hill”; [Interviewer: “Any significant changes since your working life?”] “Depression… And it’s still there.”*
(Margery, age 70, 113 min. MVPA/day)

Those who were socially isolated but not lonely (Profile C) did have close, although infrequent, contact with their families.


*“Yes, [my son] sings over the last few years… so whenever they put on a concert I think ‘yeah!’ I get to come you see… Well perhaps 3 or 4 times a year?”*
(Rose, age 68, 84 min. MVPA/day)


*“There’s the regular contact with my sister in Norwich, and um my son, that may or may not be meeting with [him], if he’s down. Because he has been down to Bath on occasions, so we’ve had meals together.”*
(Robert, age 74, 58 min. MVPA/day)

Participants who were neither socially isolated nor lonely (Profile A) tended to have a wide range of social contact with neighbours, friends and family.


*“We have family get-togethers all the time… we all meet up we go for a walk, the family”; “We’re going down to Bay at Easter. And then I shall take my kayak down there, because the children will be down there, the grandchildren now. We’ve rented, like we do, we’ve rented a big house down there so the whole family can go down.”*
(Reese, age 72, 71 min. of MVPA/day)


*“I try and do something every day. You know, Monday I do aqua fit… the other day, the friends that I have actually met at aqua, we went to [a nearby town].”; “And I help down at the, you know, the [social] club in the village.”*
(Ashley, age 69, 83 min. of MVPA/day)

##### Environmental Domain

There was little diversity in perceptions of the environment across the high-MVPA group. Most participants perceived themselves to have good or adequate access to walking paths and picturesque countryside surrounding them. All participants used a private car to get to social and active pursuits out of their own village, and all but one participant felt reliant on this form of transport.


*“You do need a car to be able to go out and do anything really.”*
(Barbara, Profile E, age 68, 66 min. MVPA/day)

#### 3.2.2. Low-MVPA Group

##### Personal Domain

Inactive participants who were not lonely or socially isolated (Profile B), and one participant who was not lonely despite being socially isolated (Profile D), differed in their level of past engagement in social activities and sense of social identity. Profile B participants had a history of high levels of social engagement and viewed themselves as having a social identity. They continued to seek social engagement in their older age, despite age-related health and functional difficulties. Those who had physical limitations managed to continue their socialising by adapting their pursuits in order to maintain the social aspects while reducing the physical demand of the activities:


*“I go to skittles but I don’t play it now... We go to the nearby village which is over that way, and it’s a damn good evening because it’s a double alley, therefore you’ve got at least 45 people there. It’s like a little party really. Cos I know most of them, so you can have a chat with anybody you know, you know, it’s a good job.”*
(Mike, Profile B, age 83, 4 min. MVPA/day)

The participant who was socially isolated but not lonely (Profile D), saw himself as never having been the “type” to engage in physically demanding pursuits. He had engaged in social pursuits, but this had been initiated by a social pressure, not an intrinsic desire for social contact.


*“I’m not really a physical type, I mean I’ve never particularly found physical exercise to be that rewarding… running round and round a block to keep fit, never been my thing.”; “First of all I was persuaded into taking part in that [cricket club] and then I decided it was quite, quite, yea not too bad, not too good but not too bad of a game and um I quite enjoyed it.”*
(Daniel, Profile D, age 68, 3 min. MVPA/day)

In contrast, participants who reported social loneliness (Profile F, Profile G) had a life-long history of active and social pursuits. However, due to a variety of personal circumstances (having moved away from previous living or work-related community, or lack of time due to family caring duties), these pursuits were disrupted which led to discontent in their level of social contact:


*“I keep wanting to establish some sort of daily schedule, some sort of daily routine. I haven’t managed that yet. Being retired nearly eight years I still haven’t managed that yet… Too busy! There’s always something else… errand or a job to do.”*
(Nathan, Profile F, age 67, 13 min. MVPA/day)


*“When I was 60 [my mother] had a massive stroke and um I’ve been looking after her full time since then. …I personally would have continued going to the flower club because I love flowers and I enjoy the socialising with the people… There are a lot of things that, yes, a lot of things that I miss out on. Because I just physically can’t do it, there isn’t time.”*
(Christina, Profile G, age 67, 11 min. MVPA/day)

Despite current physical limitations, individuals who were not lonely (Profile B and Profile D) showed high levels of satisfaction when looking back over their life’s achievements:


*“We’ve had a very good life actually. I’m not complaining about any of it, we’ve been very, very fortunate… I’m happy with it.”*
(Mary, Profile B, age 81, 2 min. MVPA/day)

Lonely participants (Profile F and Profile G), in contrast, showed a discontent with their current life, some wishing their retirement had happened differently and some wishing to be able to establish different activity patterns:


*“Of course he and I, had thought when we retired, that we’d go to the pub and go and see the countryside and drive around the countryside and go and see places we haven’t been to before. And that, we’ve failed miserably on all of that.”*
(Christina, Profile G, age 67, 11 min. MVPA/day)

##### Social Domain

Across the low-MVPA group, there seemed to be limited contact with the local neighbourhood community and there were differences in the level of contact with friends and family. Participants who were not socially isolated or lonely (Profile B) had frequent and close contact with friends and family:


*“We’ve got quite a big circle of friends, and we entertain and we go to other people’s for tea… every week we try and do something with friends.”*
(Ian, Profile B, age 87, 4 min. MVPA/day)

The one participant who was socially isolated but not lonely (Profile D) had infrequent contact with long-standing friends:


*“We have a few get-togethers now and then [with friends from previous Cricket steward team]… We used to have an annual visit somewhere.”*
(David, Profile D, age 68, 3 min. MVPA/day)

Those feeling socially lonely (Profile F; Profile G) lived far away from their friends and family and saw them infrequently due to the distance or lack of time:


*“[Friends in our previous village] were such a jolly lot. It was good. I miss them very much.”; “At the moment we haven’t been going out much because we’re always doing one of the flats [work]. It’s been a bit of a bugbear… I want to go and visit friends again. Because we’ve been so cut off from people recently, with the lifestyle that we’ve had. I would like to go and see friends again.”*
(Sandra, Profile G, age 71, 6 min. MVPA/day)

##### Environmental Domain

A common theme across the low-MVPA group was the perceived inadequacy of the environment for engaging in PA, including available facilities, a lack of pavement, agricultural land use or bad weather. Most participants with low-MVPA lived in the same rural villages as the highly active participants:


*“The traffic has increased. We’ve got bicycles in the shed but we never use them because it’s too dangerous.”*
(Ray, Profile D, age 77, 3 min. MVPA/day)


*“The road is terrible. We haven’t got a path until we get just past that mucky farm, and that’s all muddy;”*
(Joan, Profile B, age 80, 7 min. MVPA/day)


*“The other thing round here is that the farmers have ploughed up a lot of the footpaths… You can walk round the edge of the field, under sufferance, if you don’t mind walking through some nettles and brambles and this sort of thing… there’s no way I’m going to walk along a field of vegetables.”*
(Barry, age 72, Profile F: 3 min. MVPA/day)

As with the high-MVPA group, all inactive participants depended on transport, either private or the bus (one participant) to engage in their daily pursuits:


*“If we didn’t have the car and we’d have to rely on buses, we’d be restricted… if they take away my licence we shall be in trouble.”*
(Ian, Profile B, age 87, 4 min. MVPA/day)


*“If they stopped the busses then that’d be it… I suppose I’d have to move.”*
(Joan, Profile B, age 80, 7 min. MVPA/day)

## 4. Discussion

This study explored the personal, social and environmental context in which physically active and physically inactive older adults experience loneliness and/or social isolation in a rural setting. Older adults who attain a high level of MVPA may experience social isolation without feeling lonely if they have life-long interests in solitary, productive pursuits and are satisfied with infrequent family contact (Profile C). However, older adults with high levels of MVPA may also experience emotional loneliness when family relationships are disrupted, despite regular contact with friends and neighbours (Profile E). Inactive older adults can be socially isolated due to poor health and function, but not lonely due to satisfaction with irregular contact with friends and having supportive and close contact with a spouse (Profile D). However, inactive older adults can also experience social loneliness after relocating to a new neighbourhood, or lacking the time to continue social activities due to occupational or caring responsibilities (Profile F, Profile G). Supporting findings from previous studies [49] this study did not identify any particularly unique contribution of living in a rural environment in the proposed profiles. The only exception seems to be the impact of transport. Although transport is an important barrier in both urban and rural settings, due to complete lack or inadequate provision of public transport in some rural areas, older people who do not have their own transport may have greater risk of loneliness and/or social isolation than their urban counterparts.

The diversity and complexity in types and intensity of PA, and its interaction with loneliness and social isolation profiles provides insight into why quantitative studies have found no association between objectively measured PA and reported loneliness or social isolation types [23,24,25]. This strengthens the position that associations found between loneliness and low PA [17] and between social isolation and low PA in older adults [21] in quantitative studies using self-reported levels of PA are only relevant to perceived PA not actual, objectively measure PA. The findings of this study stress that it may be possible to be socially isolated or to feel lonely despite high levels of objectively measured PA that includes minutes of MVPA gained through any everyday activity, not just leisure or occupational PA [26].

Our findings add to a body of qualitative studies which, although they have explored experiences of, and precursors to, loneliness and social isolation in diverse older populations, they have not investigated the experiences of loneliness or social isolation of people with contrasting groups of PA level [14,50,51,52,53,54]. The findings support qualitative literature regarding the precursors of loneliness, such as a lack of intimate relationships [55,56,57,58] and informal caring roles [59]. They also support commonly cited reasons for preventing loneliness such as good health/function, a history of PA, social motivation, and access to transport [60,61].

The perspective of PA not being associated with loneliness is novel in qualitative research. and stressed the importance of revisiting the notion of lonely older individuals being less physically active and vice versa [20,62].

Our findings highlight the need to be cautious about supporting stereotypes of older age, as these could reduce the effectiveness of therapeutic interventions and perpetuate negative societal views about ageing [40,63]. Assuming that healthy, active older people will not feel lonely could lead to missed opportunities to assist those that do feel lonely due to issues such as intimate/family relationship disruption. In reverse, the stereotype that inactivity and loneliness naturally coincide may lead to perpetuating negative ageist attitudes regarding age-related physical changes, inactivity and the onset of loneliness [64]. Encouraging negative ageing attitudes in older adults can have harmful effects on older individuals’ self-evaluation and behaviour, and may even lead to feelings of loneliness [40,65]. The diversity in our findings also highlights the need for highly individualised approaches to alleviate loneliness in older adults.

These findings support of the use of both the cognitive theory of loneliness [7] the deficit theory of loneliness [8]. Regardless of PA level, participants tended to continue the pursuits (active, social or both) in which they had engaged previously in life, making adjustments to activities to become less physically demanding if age-related barriers to physical exertion were present. Only when the expected and desired level of social contact was disrupted, did participants experience feelings of either emotional loneliness or social loneliness. Therefore, the presence of loneliness can be explained by both the characteristics of people’s network of relationships (the deficit theory of loneliness) and their relationship preferences (the cognitive theory of loneliness) [57]. Contrasting loneliness classifications between the questionnaire responses and the interviews regarding “social loneliness”, could indicate the limited ability of a single-item question about loneliness to capture experiences of social loneliness in older people.

The hypothesis that lonely individuals tend to withdraw from social interactions and thereby become less physically active [62] was not supported by the findings. Profile E included participants who were lonely but highly physically active and had frequent local and further social interactions. For these participants, the deficit theory [8] highlighting the impact of the absence of a close family relationship seems to better describe their loneliness. The deficit theory was also supported by the profiles of inactive, but lonely participants (Profiles F and G), for whom the absence of frequent contact with friends was linked with reports of loneliness.

The current study explored social isolation and loneliness using the adapted-socioecological framework adding knowledge regarding the wider applicability of this framework to social behaviours and social experiences. The socioecological model was developed to describe the determinants of any health-related behaviour or states [37]. Its current extension to the context associated with social isolation or loneliness is therefore consistent with its original design given that both constructs are recognised as health-limiting states [2]. Both the cognitive and deficit theories of loneliness present a good fit within the personal and social domains of the adapted socioecological model. The use of this model also allowed the observation of a theoretical extension of the cognitive theory: the influence of social pursuits contacted earlier in the life-course on one’s expectations of social relationships in later life.

The findings of this study highlight the complexity of the relationship between PA and social isolation and/or loneliness such that increasing PA may not result in some people experiencing less loneliness or isolation. Although PA interventions could provide multiple benefits they should not be seen as a standalone intervention targeting older people who are isolated or lonely, but may be considered as part of a multifactorial response tailored to individual’s needs.

### 4.1. Strengths and Limitations

This study shed light into how physically active and inactive older adults can be susceptible to social isolation and/or loneliness. However, these findings are exploratory in nature and in need of further verification. The geographical confinement to one south-western county in England limits the socio-demographic diversity of the sample and the transferability of findings to other rural locations in the UK. Further, while we sampled from 20 rural villages across South West England, aiming to capture social and geographical diversity, the sample was exclusively white-Caucasian and, on average, highly educated and affluent. This leads to limited generalisability to rural areas with ethnically diverse populations and high socio-economic deprivation.

A strength was the strategic matching of older people in high and low groups of objectively measured PA, keeping gender, age and functional ability score as comparable between groups as possible. However, selecting the highest and lowest quintiles of objectively measured MVPA limited the analysis in gaining an insight into the diversity of the experiences of older adults with a middle-range level of MVPA.

There was a three-to-six-month gap between the accelerometer measurements during the quantitative phase of SHARP and the qualitative interviews which might have resulted in changes to PA level by the time of interviews. This was compensated for by reviewing the participant’s seven-day diary (obtained during the same week as the accelerometer data) in the interview and asking if and how any activities had changed. Lastly, while the interviews provided in-depth information about social activities, experiences and wellbeing, direct questions about loneliness were not asked due to ethical concerns. As a result, the interviews may not have captured all instances of loneliness.

### 4.2. Research Recommendations

Future qualitative work should employ a more ethnically and geographically diverse sample to further develop the notion of different PA, social isolation and loneliness profiles. A worthwhile addition to the mixed-methods used in this study would be the survey measurement of both the social and emotional loneliness concepts from the deficit theory [8] and using quantitative and qualitative methods within a closer time frame. Given the strong emergence of the influence of earlier life experiences on present-day expectations and social/active engagement, the use of a life-course perspective in a further qualitative enquiry is also recommended.

The current study cannot determine what might happen if socially isolated or lonely older adults were to increase their PA level. Although a large number of published studies in physical activity loneliness and social isolation in later life rely on self-reported PA measures, a growing literature on active ageing employs objective measurement of physical activity. Future studies should incorporate social isolation and loneliness as outcome measures in order to evaluate how these aspects change with increased PA [66,67].

## 5. Conclusions

This study found wide diversity in personal, social and environmental contexts related to loneliness and social isolation in highly active and inactive rural-living older adults. It is possible for highly active older people to be socially isolated (but not lonely) if their interests lie in solitary pursuits, and possible for them to experience emotional loneliness (but not social isolation) when intimate relationships are disrupted. It is possible for inactive older people to adapt their activities to meet their expectations of social contact and, in so doing, to avoid becoming socially isolated and/or lonely. This diversity highlights the need to reconsider stereotypes of lonely and non-lonely older people and to identify individualised approaches to alleviate loneliness in later life. Increasing PA opportunities as a standalone strategy may not be sufficient to address loneliness or isolation. However, the promotion of a healthier and more active lifestyle via multifactorial approaches that specifically target social isolation and/or loneliness could contribute to better well-being and maintenance of good health in later life.

## Figures and Tables

**Figure 1 ijerph-18-03971-f001:**
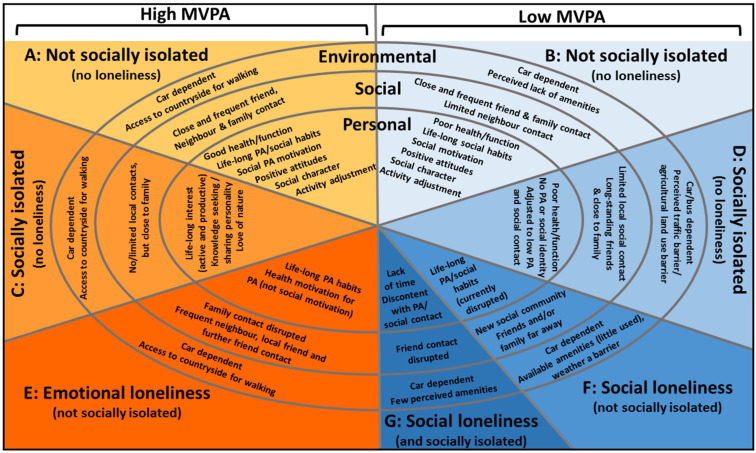
Socioecological characteristics observed in different MVPA, social isolation and loneliness profiles. The personal domain contains both characteristics relating to the present time (e.g., health) and previous life-stages (e.g., life-long habits of PA). Themes affecting high-MVPA participants are displayed on the left, and those affecting low-MVPA participants on the right side of the figure.

**Table 1 ijerph-18-03971-t001:** Example interview statements for classifying social isolation and loneliness.

**Example Statements**	
Social isolation	*“I hardly ever see XXX”* *“I only see XXX about twice a year”*
**Discrepancy Theory of Loneliness**	
Emotional loneliness	*“I miss a really close relationship…”* *“I don’t feel close to anyone…”*
Social loneliness	*“I miss seeing my old friends/neighbours…”*
**Cognitive Theory of Loneliness**	
Dissatisfaction with levels of contact	*“I don’t have enough contact with my family/friends/neighbours…”* *“In my retirement I expected to have more contact with my friends…”* *“I get very sad about not seeing my family…”*

**Table 2 ijerph-18-03971-t002:** Participant characteristics in the low- and high-moderate-to-vigorous physical activity (MVPA) groups.

Pseudonym (Sex)	Age	MVPA	PF ^1^	Interview Data		Questionnaire Data	
				Loneliness ^2^	Social Isolation	Loneliness	Social Isolation
**High MVPA**							
Phil (M)	66	89	12	Not lonely	Not SI	Hardly ever	Not SI
Mark (M)	66	87	11	Not lonely	Not SI	Hardly ever	SI family
Rose (F)	68	84	9	Not lonely	SI all	Hardly ever	SI all
Bill (M)	68	81	10	Not lonely	Not SI	Hardly ever	SI family
Barbara (F)	68	66	10	E. lonely	Not SI	Often	SI family
Ashley (F)	69	83	11	Not lonely	Not SI	Sometimes	SI family
Margery (F)	70	113	11	E. lonely	Not SI	Sometimes	SI family
Vanessa (F)	71	65	11	Not lonely	Not SI	Hardly ever	Not SI
Reese (M)	72	71	10	Not lonely	Not SI	Hardly ever	SI family
Robert (M)	74	58	9	Not lonely	SI friends and neighbours	Hardly ever	SI family and friends
Warren (M)	77	39	10	Not lonely	Not SI	Hardly ever	SI family
Isla (F)	81	40	10	Not lonely	Not SI	Hardly ever	SI friends
**Low MVPA**							
Nathan (M)	67	13	12	S. lonely	Not SI	Hardly ever	SI family
Christina (F)	67	11	10	S. lonely	SI friends	Hardly ever	SI family
Daniel (M)	68	3	7	Not lonely	SI friends	Hardly ever	SI family and friends
Sandra (F)	71	6	11	S. lonely	SI friends and neighbours	Hardly ever	SI family and friends
Barry (M)	72	3	6	S. lonely	Not SI	Hardly ever	SI friends
Janice (F)	73	4	6	Not lonely	Not SI	Hardly ever	SI family
Eve (F)	77	4	8	Not lonely	Not SI	Hardly ever	SI family
Ray (M)	77	3	11	Not lonely	Not SI	Often	SI family
Joan (F)	80	7	6	Not lonely	Not SI	Hardly ever	SI friends and neighbours
Mary (F)	81	2	9	Not lonely	Not SI	Hardly ever	SI family
Mike (M)	83	4	9	Not lonely	Not SI	Hardly ever	Not SI
Ian (M)	87	4	9	Not lonely	Not SI	Hardly ever	SI family

^1^ Physical function score out of 12; ^2^ E. lonely = emotionally lonely, S. lonely = socially lonely. Underlined = Loneliness observed in interview data.

**Table 3 ijerph-18-03971-t003:** Seven profiles of moderate-to-vigorous physical activity (MVPA), social isolation and loneliness.

	High MVPA	Low MVPA
**Not Socially Isolated**	**Profile A:** Not socially isolated/ not lonely (8/12)	**Profile B:** Not socially isolated/ not lonely (7/12)
	**Profile C:** Socially isolated/ not lonely (2/12)	**Profile D:** Socially isolated/ not lonely (1/12)
**Lonely**	**Profile E:** Emotional loneliness/ not socially isolated (2/12)	**Profile F:** Social loneliness/ not socially isolated (2/12)
		**Profile G:** Social loneliness/ socially isolated (2/12)

## Data Availability

Data supporting reported results can be obtained via communication with corresponding author.

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
