# Peer review of "Profiles of Loneliness and Social Isolation in Physically Active and Inactive Older Adults in Rural England"

_ijerph, 2021, doi:10.3390/ijerph18083971_

Round 1
Reviewer 1 Report
The presented manuscript is very interesting and highly relevant. It is worth paying attention to the processes of social isolation of older people, especially those living in rural contexts.
The paper is also very well written and structured.
Below we point out some issues that could improve the submitted manuscript:
- It would be interesting if the authors included the interview script used as a table or as appended material.
Author Response
Please see the attached file. We provide point by point response to comments/suggestions made by all reviewers.

Reviewer 2 Report
Estimated, The theme that his article deals with is relevant, taking into account the current situation, that is, older people with high rates of loneliness and older people both physically active and inactive in a rural setting, which is very interesting. He indicates in the abstract that loneliness leads to mortality in old age, and I have been curious about what effects loneliness has to say that it leads to mortality. Suggestions for improvement linked to your objective: Show in the introduction clearly or clarify it better if there are studies on this objective and if there are differences. In the discussion really clarify the differences between active and inactive, because, I get the impression, that both are the same, and that in this article being active or inactive would not influence loneliness / isolation. And that, therefore, a final conclusion would be that their study indicates that there are no differences between active and inactive people and their influence on loneliness, that is, that the practice of physical activity does not influence greater relationships or loneliness / isolation. Please, explain better in a more forceful way what you want to say in your article.Author Response
Please see the attached document. We provide point by point response to comments/suggestions by all reviewers.

Reviewer 3 Report
Thank you for inviting me to review this interesting paper reporting qualitative data from the previous reported SHARP study - the paper is interesting but can be improved as follows
Please define PA early in the introduction. There are many references to the original sharp study but to make sense of the current study, far greater information from SHARP should be provided regarding the aims and methodology , methods of recruitment and tools used in quantitative part of the study to measure loneliness for example. It is very difficult to make sense of all the findings without reverting to the original papers and this needs addressing before publication. We are told participants were recruited from 20 villages , but the previous SHARP papers report recruitment from 23 villages ?
The authors mention that participants were contacted by phone to participate – were they sent information regarding the study and were they given time to decide whether they wish to participate? The paper states the age range was 66- 83 years however in Table 2 the oldest participant is 87 years. Nine participants are under 70 and only five were over 80 – how does this compare with the sample of the original SHARP study as I would argue this research is based on “young” older people and that is not discussed within the paper – a great age range would probably have yielded different results. The fact that all participants in the qualitative study had access to a car is partly due to fact these were not the very elderly but more importantly the issue of Socioeconomic deprivation is not discussed and that is an important omission which needs to be addressed.
Author Response
Please see the attached document. We provide point by point response to comments/suggestions by all reviewers.
Round 2
Reviewer 2 Report
After reading, all changes have been incorporated and it is fit for publication.